# Monte Carlo Tree Descent for Black-Box Optimization

**Yaoguang Zhai**
UCSD

**Sicun Gao**
UCSD

## Abstract

The key to Black-Box Optimization is to efficiently search through input regions with potentially widely-varying numerical properties, to achieve low-regret descent and fast progress toward the optima. Monte Carlo Tree Search (MCTS) methods have recently been introduced to improve Bayesian optimization by computing better partitioning of the search space that balances exploration and exploitation. Extending this promising framework, we study how to further integrate sample-based descent for faster optimization. We design novel ways of expanding Monte Carlo search trees, with new descent methods at vertices that incorporate stochastic search and Gaussian Processes. We propose the corresponding rules for balancing progress and uncertainty, branch selection, tree expansion, and backpropagation. The designed search process puts more emphasis on sampling for faster descent and uses localized Gaussian Processes as auxiliary metrics for both exploitation and exploration. We show empirically that the proposed algorithms can outperform state-of-the-art methods on many challenging benchmark problems.

## 1 Introduction

Black-Box Optimization (BBO), also referred to as Derivative-free or Zeroth-order Optimization, considers objective functions that are not known analytically and can only be evaluated at various inputs, potentially at a high cost. The generality of the formulation makes BBO broadly applicable to a wide range of challenging problems in machine learning [1, 2, 3] as well as many scientific and engineering problems [4, 5, 6]. BBO problems over compact domains are naturally NP-hard: in the worst case, we need to exhaustively search through the combinatorially-large number of local regions to find high-quality solutions. Thus, the goal of BBO algorithm design is to accelerate optimization progress with respect to the number of function evaluations.

Existing work on BBO can be categorized into model-based and model-free approaches. Most model-based approaches, typically in the framework of Bayesian Optimization [7, 8], involve learning a surrogate function from samples of the unknown function and optimizing the surrogate rather than the original function. For highly nonlinear functions with high-dimensional input spaces, such methods are known to be costly because of the need for global modeling of the objective functions. Various Bayesian optimization approaches utilize ensembles of local surrogate models [9] to improve performance. Model-free approaches include simulated annealing [10], cross-entropy methods [11], search gradient [12], as well as traditional direct search methods such as Nelder-Mead [13, 14]. The goal is to iteratively propose sampling distributions that can approach the optima. Such methods typically do not attempt to maintain global information about the objective and are challenged when the optimization landscape is highly non-convex [15]. In general, the lack of mechanisms for explicitly managing the search over the combinatorially-large number of local regions, in both standard model-based and model-free BBO methods, has been a major bottleneck of the field.

Recent advances in stochastic tree search methods [16, 17] offer new opportunities for balancing local search and modeling with more systematic global exploration in BBO problems. In particular, Monte Carlo Tree Search (MCTS) has recently been introduced for computing good partitioning of the search space for BBO [3, 18, 19]. These approaches adaptively divide the input space into regions,

36th Conference on Neural Information Processing Systems (NeurIPS 2022).

balancing exploitation and exploration, to only perform Bayesian optimization at local regions and create better model ensembles. However, because the focus is still on modeling the objective, the ability of MCTS to quickly expand deep branches into promising search regions has not been fully utilized. As a result, the curse of dimensionality can still quickly stall the search, while model-free descent methods may be able to make more progress if they are also guided by MCTS.

We propose a new design of MCTS methods for BBO, with more emphasis on sample-efficient local descent, which can benefit the most from balanced exploitation and exploration. We use Bayesian optimization and local modeling as auxiliary metrics for guiding the search tree construction. At each node in our search tree, we iteratively collect samples in the neighborhood of some anchor point and also maintain a local Gaussian Process (GP) model for the neighborhood. The samples are chosen using sampling-based descent such as Stochastic Three Points methods (STP) [20], and they are also used to train the local GP models. These local models provide surrogate objectives to propose future samples without querying the ground truth function, and they also provide uncertainty metrics for exploration steps. We name our overall approach Monte Carlo Tree Descent (MCTD), because of the focus on faster descent led by samples that are managed by tree search, rather than using MCTS for explicit space partitioning. We evaluate the proposed methods with experiments on challenging benchmarks such as nonlinear optimization benchmarks [21], policy search for MuJoCo locomotion tasks [22], and neural architecture search [23]. We compare our algorithm with state-of-the-art model-based [9] and MCTS-based methods [19], as well as model-free [24] and direct search methods [13]. We observe clear benefits in the proposed designs for improving efficiency, consistently outperforming existing methods on the tested benchmarks.

## 2   Related Work

**Model-based methods.**   Bayesian optimization [7, 25] typically uses Gaussian Processes to construct surrogate models of the objective functions [8], with samples selected by acquisition functions (e.g., confidence bounds, expected improvement, etc.) [26, 27]. Model-based methods are known to suffer from the curse of dimensionality as the problem dimensionality and sample sizes grow quickly [28]. Many approaches have been proposed to improve the scalability of Bayesian optimization methods in high-dimensional problems [29, 30, 31]. For instance, TuRBO is a state-of-the-art method that uses Thompson sampling with Expected Hypervolume Improvement (EHVI) [9]. It samples in local trust regions and adjusts the trust regions after each sampling iteration, which has shown major benefits in improving the efficiency of model-based approaches for BBO.

**Model-free methods.**   Model-free approaches focus on sampling inputs, either point-wise or population-based, that can incrementally approach optimal regions in the search space without explicitly maintaining models of the objective. Standard approaches include stochastic methods such as simulated annealing (SA) [10] and cross-entropy (CE) [11] and deterministic schemes such as Nelder-Mead (NM) [13]. These methods have been successfully applied to a wide range of problems but they typically do not aim for optimizing efficiency, i.e., reducing the number of evaluations [32]. They may still offer improvements faster than local methods that rely on gradient information [14, 32]. The Stochastic Three Points method [20] is a simple but effective way of direct search that compares function values at the base point, in one random direction, and in the opposite direction. Each step evaluates only two more points that lie in the opposite direction of the current point and moves towards the one with a better value. To improve sample efficiency, we attempt to combine The Stochastic Three Points method with model-based methods and carefully design the direction in which the method will try in each iteration.

**Tree search methods.**   Various tree-search methods have been proposed to improve partitioning of the search space in BBO, such as Deterministic and Simultaneous Optimistic Optimization (DOO and SOO) [18], and Hierarchical Optimistic Optimization (HOO) in [33]. Specifically, DOO divides up the search domain into partitions, each of which is represented by a point within it, assuming known Lipschitz constants for the objective function. SOO and HOO extend DDO to stochastic versions but are mostly applicable to low-dimensional problems because of the high cost involved in creating good partition cells. Voronoi Optimistic Optimization (VOO) [3] can be more efficient in high dimensions by combining Voronoi partitioning and tree search. LA-MCTS [19] introduces MCTS to manage the partitioning of the search space. It learns latent actions that define boundaries between good and bad regions in the search space and prioritizes the expansion of the search tree

around the boundaries. When continuing with such splitting, it sets a sampling preferential on every node in the tree. In every iteration, the search tree is traversed from the root node to a leaf by selecting the highest approximated value based on Upper Confidence bounds applied to Trees (UCT) algorithm. The optimization is then performed from the subspace partition on the selected node. These methods successfully change the objective function modeling for global space to local regions. However, the partitioning of the state space, particularly when the space is a high dimension, becomes a very challenging problem. The tree becomes extremely large when the optimization attempts to learn with high accuracy in local regions.

## 3   Preliminaries

We consider the problem of minimizing an objective function $f(x) : \Omega \to \mathbb{R}$ where the domain $\Omega \subseteq \mathbb{R}^n$ is compact. We assume the ability to evaluate $f(x)$ for arbitrary $x \in \Omega$ but do not have information about the analytic form of the function or its derivatives.

Gaussian Processes (GP) is commonly used in Bayesian optimization and is also used in our work to construct a surrogate model for the local model-based optimization. For a finite collection of points $x_1, ..., x_k \in R^d$, GP constructs the mean vector $\mu_0$ from the function $f$ at each $x_i$, and the covariance matrix $\Sigma_0$ by a kernel at each pair of $(x_i, x_j), i, j = 1, 2, ...k$. With $\mu_0$ and $\Sigma_0$ the prior distribution on $f$ is:

$$f(x_1, \dots x_k) \sim \mathcal{N}(\mu_0(x_1, \dots x_k), \Sigma_0(x_1 \dots x_k; x_1, \dots x_k)) \tag{1}$$

For any new point $x$, we can use Bayes' rule to compute the conditional distribution of $f(x)$:

$$f(x|x_1, \dots x_k) \sim \mathcal{N}(\mu_0(x_1, \dots x_k, x), \Sigma_0(x_1, \dots x_k, x; x_1, \dots x_k, x)) \tag{2}$$

The Stochastic Three Points (STP) method is a model-free approach to BBO that uses only a small number of samples in each iteration to identify descent directions. At each time step $t$ with a current sample $x_t$, it generates a set $D_t = \{x_t, x_t + s_t \cdot \alpha_t, x_t - s_t \cdot \alpha_t\}$ where $s_t$ is a direction and $\alpha_t > 0$ is the step size at step $t$. When $\alpha_t$ is small enough, the relationship between $f(x_t + s_t \cdot \alpha_t)$, $f(x_t)$ and $f(x_t - s_t \cdot \alpha_t)$ is monotonically non-increasing or non-decreasing if the gradient of the function $f$ is not zero in the direction of $s_t$. For the next step, $x_{t+1} = \operatorname{argmin}_{x \in D_t} f(x)$. In our method, the STP-based local descent optimization will identify the best direction $s_t$ with an optimized step size $\alpha_t$ for improving its performance.

Monte Carlo Tree Search (MCTS) is a leading framework for balancing exploration and exploitation in sampling-based tree search. It consists of four main steps: Selection, Expansion, Simulation, and Backpropagation. During Selection, a search tree is traversed from the root node to a leaf node. This traversal is made by selecting the node with the highest value based on the UCT algorithm. For a node $n_i$, the UCT $\nu$ is computed by:

$$\nu(n_i) = R_i/N_i + C \cdot \sqrt{2 \cdot \ln N_b/N_i} \tag{3}$$

in which $R_i$ is the rewards on $n_i$; $N_i$ and $N_b$ denote the number of visits on $n_i$ and its parent node $n_b$, respectively; $C$ is a constant to balance between exploitation and exploration. At each branch node $n_b$, the child to select is the one with the highest $\nu$ value among all of its immediate children. At Expansion, a new child node is then added to expand the tree. During Simulation, a random simulation is run from the new child node until the terminal node is reached, and the simulation reward is approximated. Finally, the simulation reward is backpropagated through the selected nodes to update the tree. In our approach, we construct our Monte Carlo tree by assigning every leaf node to one optimization process. During each step, we use a modified UCT algorithm to select the node on which the optimization is launched.

## 4   Monte Carlo Tree Descent

Our MCTD algorithm iteratively constructs a search tree over the domain of the objective function, and at each node of the tree we maintain a set of samples and a surrogate model learned from them. The balancing of exploration and exploitation takes into account several factors that will be explained in the subsequent sections. The overall algorithm is illustrated in Alg.1, and we refer Fig. 4 in the Appendix that provides a visual illustration of the process.

---

**Algorithm 1** Monte Carlo Tree Descent (MCTD)

 1: **function** MCTD(objective: $f$, domain: $\Omega$)
 2:     $x \leftarrow$ random sample in $\Omega$
 3:     $n_0$.sample_set $\leftarrow (\mathbf{x}, \mathbf{y})$    ▷ root node
 4:     **for** step $= 1, ..., t$ **do**
 5:         $n \leftarrow \text{Select}(n_0)$
 6:         Optimize$(n)$
 7:         Backup$(n)$
 8:     **end for**
 9:     **return** $y_0^*$
10: **end function**
11:
12: **function** EXPAND(node: $n_i$)
13:     **if** $n_i$ is leaf **then**
14:         $n_{i0} \leftarrow n_i$ excluding parent/child
15:         $n_i$ child list $\leftarrow n_{i0}$
16:     **end if**
17:     $lv \leftarrow$ level of $n_i$
18:     $d \propto \exp(-lv)$
19:     $x \leftarrow$ random sample in $B(x_i^*, d)$
20:     $n_{im}$.sample_set$\leftarrow \{(x, f(x))\};$
21:     $n_i$.children.append$(n_{im})$
22:     **return** $n_{im}$
23: **end function**
24:
25: **function** BACKUP(node: $n_i$)
26:     $n \leftarrow n_i$
27:     **while** $n$ has parent $n_p$ **do**
28:         Update $(x_p^*, y_p^*)$ with Eq.5
29:         Update $dy_p$ with Eq.6
30:         $n \leftarrow n_p$
31:     **end while**
32: **end function**

 1: **function** SELECT(node: $n_i$)
 2:     $n_b \leftarrow n_i$
 3:     **while** $n_b$ has children **do**
 4:         **for** child node $n_{bi}$ **do**
 5:             Compute $\nu(n_{bi})$ by Eq.4
 6:         **end for**
 7:         Compute $\nu(n_{bx})$ by Eq.7
 8:         **if** $\max_i(\nu(n_{bi})) < \nu(n_{bx})$ **then**
 9:             **return** Expand$(n_b)$
10:         **end if**
11:         $\hat{b} \leftarrow \operatorname{argmax}_i \nu(n_{bi})$
12:         $n_b \leftarrow n_{b,\hat{b}}$
13:     **end while**
14:     **if** $EP(n_b)$ in (8) is satisfied **then**
15:         $n_b \leftarrow$ Expand$(n_b)$
16:     **end if**
17:     **return** $n_b$
18: **end function**
19:
20: **function** OPTIMIZE(node: $n_i$)
21:     $\alpha_D \leftarrow 1$
22:     **if** $|n_i.\text{sample\_set}| >= NR$ **then**
23:         $\Theta \leftarrow$ GP model of $n_i$.sample_set
24:         $\alpha_D \leftarrow \alpha_D \cdot$ correlation length in $\Theta$
25:     **else**
26:         oracle $\Theta \leftarrow$ None
27:     **end if**
28:     Descend on $n_i$ by $\Theta$, $f$, $\alpha_D$ from $(x_i^*, y_i^*)$
29:     $n_i \leftarrow$ Bayesian Optimize from $\{(x, y)\}_i$
30:     Update $(x_i^*, y_i^*)$ and $dy_i$ by Eq.5 and 6
31:     **return**
32: **end function**

---

## 4.1   Overall Tree Search Strategy

We initialize our algorithm at a random sample in the domain of the objective function, and the sampled points create the root node of the entire search tree. Unlike standard MCTS that considers finite and discrete actions at each node, for BBO over the continuous domains we can not expand the infinitely-many possible next samples as child nodes of the root node. Consequently, already at the root node, we need to decide between two choices. First, we could perform local descent on the current sample at this node. Second, we could explore a different region in the space by taking a sample that is far from the current one, which will act as a new anchor point that forms a new child node of the tree, which expands the tree. When multiple child nodes have been expanded at a node, there is the third option of going down the tree along the most promising branch, and then focusing the next steps of search from there.

Consequently, in each iteration of the algorithm, we perform three operations sequentially. First, we perform branch selection starting from the root node, and then either land at some existing node or create a new anchor sample and node, from which we will perform local descent.

## 4.2   Branch Selection

In every step, we pick a leaf for optimization. To balance exploration and exploitation, our algorithm uses UCT to determine the path between the root and the leaf, as shown in the function **SELECT** in Alg.1 line 1. We modified the UCT formula for fitting our MCTD algorithm. For each child node $n_{bi}$

with the parent node $n_b$ , its UCT $\nu(n_{bi})$ is given by:

$$\nu(n_{bi}) = -y_{bi}^* + C_d \cdot \sum_{j=1}^{J} dy_{bi}^{-j} + C_p \cdot \sqrt{\log N_b / N_{bi}} \tag{4}$$

Here, $C_d$ is a weight factor controlling the importance of recent improvements during optimization, $C_p$ is a hyper-parameter for the extent of exploration, $N_b$ and $N_{bi}$ are the number of visits to the branch node $n_b$ and the child node $n_{bi}$, respectively. $y_{bi}^*$ is the current best function value in the sample set $\mathbf{S}_{bi} = \{(x, y)\}$ which stores the samples during optimization on node $n_{bi}$:

$$(x_{bi}^*, y_{bi}^*) = \underset{y}{\mathrm{argmin}}(x, y), (x, y) \in \mathbf{S}_{bi} \tag{5}$$

and $dy_{bi}^{-j}$ is the most recent $j$'s improvement at $n_{bi}$ after calling the objective function. When computing $\nu$, only the last $J$ improvements are taken into account. For every call to the objective function during the optimization, we record the improvement in the function value from this call. If the value from this call is worse than the optimal value before the call, we set the improvement to zero; otherwise, we set the improvement as the absolute difference between the optimal value before and after the call. That is, for $y_{bi}^*$ at the time step $t$ as $y_{bi}^*(t)$,

$$dy_{bi}^{-j}(t) = \max(y_{bi}^*(t-j) - y_{bi}^*(t-j+1), 0) \tag{6}$$

We similarly integrate the tree expansion as the UCT algorithm. At a branch node $n_b$, in addition to examining the UCT of all its child nodes we add an artificial exploration node $n_{bx}$ that has the UCT value $\nu(n_{bx})$ as following:

$$\nu(n_{bx}) = -\sum_i (y_{bi}^*)/D_b + C_p' \cdot \sqrt{\log N_b} \tag{7}$$

where $D_b$ is the number of children of the node $n_b$, $C_p'$ is a hyper-parameter for the extent of exploration but may be different from $C_p$. This exploration node is to determine whether to optimize in a new domain because the existing children are not performing well enough. When the exploration node is selected, a new child node under the branch node is created and returned.

If the path selects a leaf that is not newly created, we need to determine whether it is worth optimizing on it. On a leaf node $n_f$, we expand the tree if the following condition is met:

$$EP(n_f): -y_f^* + C_d'' \cdot \sum_{j=1}^{J''} dy_f^{-j} < C_p'' \cdot \sqrt{\log N_f} \tag{8}$$

Here, $C_d''$ is a weight factor for recent last $J''$ improvements and may be different from $C_d$, $C_p''$ is also a hyper-parameter for the extent of exploration different from $C_p$ and $C_p'$. In the event the condition 8 is met, we will make a leaf expansion; otherwise, we descend on the selected leaf node $n_f$.

### 4.3 Tree Expansion

When we need to take an exploration step at a node, a new child node will be created. The new child node is created at a random point lying within some distance from the selected node. The minimum and maximum distances are set to $10\%$ and $50\%$ of the domain's dimensional length, with exponential decay according to the node level. After the newly created child node is placed, it will be immediately selected as the node for optimization at the current step. When the selected node to explore is a leaf node $n_f$, a new child node $n_{f1}$ is created in the same way as above, making $n_f$ a branch node. At this time, a new node $n_{f0}$, starting from the current best point at $x_f^*$, is also created as the child 0 of node $n_f$. This node $n_{f0}$ inherits a batch of samples that are near its starting point $x_f^*$, as well as the latest improvement history on $n_f$. The reduced number of samples forces the inheriting node $n_{f0}$ to focus on optimizing in the neighborhood of the starting point, while the newly expanded node $n_{f1}$ is optimizing in a distant region. Thus, the tree grows a leaf node $n_{f1}$ while maintaining the possibility of further exploiting around the best point found on $n_f$ at node $n_{f0}$. These steps are in the function **EXPAND** in Alg. 1 line 12, and 3 subplots in Fig.4 show an example. As in Fig.4 (c), the expansion takes place on the root node $n_0$. The node $n_{01}$ is a new node for exploration, placed distant from $n_0$. Node $n_{00}$ starts from $x_0^*$. Similarly, in 4 (d), node $n_{010}$ starts from $x_{01}^*$, and node $n_{011}$ is placed away from $n_{01}$, but the distance between node $n_{01}$ and $n_{010}$ is much smaller than the distance between node $n_0$ and $n_{01}$ at node creation. Fig. 4 (e) shows how a new leaf node is created.

### 4.4 Local Optimization.

In every iteration, we use the STP method to attempt local descent and also use TuRBO-1 [9] for local Bayesian optimization (BO). We tightly integrate the two methods. Samples obtained from local descent optimization are used to construct the surrogate GP regression model. The GP model not only serves as an oracle for the local descent optimization but also provides the correlation length according to which the local descent optimization scales its step sizes.

**Local Descent.** We use the STP method with the following changes. In STP, the direction $s_t$ at step $t$ is usually selected from a sphere with uniform distribution in direct search. Instead, we use the surrogate GP regression model to identify the point with the highest expected improvement. The steps of local descent optimization are as follows:

1. Choose a node $n_i$ by **SELECT**. If the number of samples exceeds some threshold, we train a Gaussian Process model that will be referred to as the oracle for this node.

2. Compute the step size $\alpha_t$. In our case, we set $\alpha_t$ to be inversely proportional to the square root of the product of node visits $N_i$ and the node level in the tree. We also rescale it according to the correlation length in the surrogate GP model when possible.

3. If the oracle is not available, get a random direction $s_t$, and use $s_t \cdot \alpha_t$ for checking ground truth.

4. If the oracle is available, generate multiple samples in the box with edge length equaling the step size $\alpha_t$, and choose the best point. The direction to the best point is $s_t \cdot \alpha_t$.

5. Start one step of STP with the selected direction of $s_t \cdot \alpha_t$ by calling the objective function.

6. Depending on the optimization progress, we may further optimize the objective function along the same direction with tuned step sizes in a fine-grain descent approach.

The last step is used when the optimization comes to fine-tuned phase with small variations in samples, so one can set a function threshold from which the search applies the fine-grain descent approach.

**Local Bayesian Optimization.** The TuRBO-1 [9] creates a hyper-rectangle Trust Region (TR) with volume $L^N$ centered at the best sample. Afterward, it samples new candidates within the TR and queries the objective function for ground truth data. The length of $L_i$ will either increase after successive "successes" or decrease after consecutive "failures". We changes TuRBO-1 in three ways to fit it into our algorithm: 1) TuRBO-1 begins with collected samples of the node. Consequently, TuRBO-1 is compelled to optimize from the vicinity of the collected sample. 2) The trust region length has been preserved on the same node, so the local BO can continue from the previous epoch. 3) We do not perform restarts for TuRBO-1 in order to avoid TuRBO-1 restarting from random samples.

### 4.5 Back Up

In the **BACKUP** function, we backpropagate the updated best score found at a leaf node and propagate it upwards to its parent nodes. This score update is important for informing future branch selections. This backup procedure is used in every step even if the best-found sample on the selected leaf node does not change after one iteration.

## 5 Experiments and Evaluation

### 5.1 Experiment Setup

**Benchmarks** We use several standard benchmark sets for testing BBO algorithms, from three categories: synthetic functions for nonlinear optimization, reinforcement learning problems in MuJoCo locomotion environments, and optimization problems in Neural Architecture Search (NAS). Synthetic functions are widely-used in nonlinear optimization benchmarks [21]. These functions usually have numerous local minima, valleys, and ridges in their landscapes which is hard for normal optimization algorithms. MuJoCo locomotion environments [22] are popular for reinforcement learning tasks. NAS problems have practical significance, since many fields are using deep learning models, but implementing efficient neural networks requires a substantial amount of time and effort.

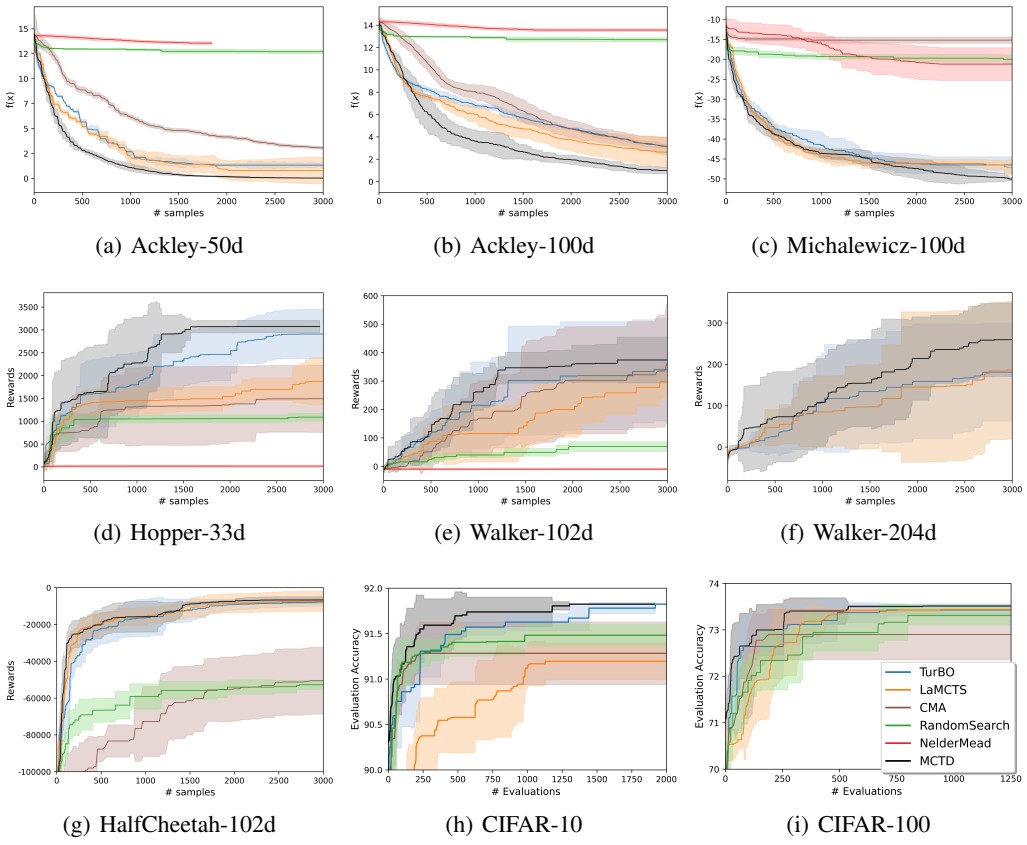

Figure 1: Overall performance of the baselines and our method. For Ackley and Michalewicz in (a), (b), and (c), the goal is to optimize for the lowest function values; in MuJoCo tasks (d), (e), (f), and (g), we aim to maximize the rewards; and for CIFAR-10 in (h) and CIFAR-100 in (i) we want to find the architecture with the highest accuracy as quickly as possible

We select multiple problems from each set, and their input dimensions range from 33d to 204d. Details of benchmark problems can be found in section B in the appendix.

**Baselines**   We selected TuRBO [9] as one baseline from the BO algorithms. La-MCTS [19] is chosen as a major comparator since this algorithm also constructs trees in a similar manner. Moreover, CMA-ES [24] from the Evolutionary Algorithm category, Nelder-Mead [13] from Direct Search algorithms, as well as the Random Search algorithm are selected for comparison as baselines.

For CMA optimization, **fmin2** from the CMA-ES package [24] is used with its default parameters. We implement our own version of the Nelder-Mead algorithm as in [13], and set its expansion coefficient, contraction inside the simplex, contraction outside the simplex, and shrink coefficient as $2.0, 0.5, 0.5,$ and $0.5$, respectively. TuRBO [9] is initialized with 20 random samples selected using Latin Hypercube sampling, and its Automatic Relevance Determination (ARD) is set to **True**. For La-MCTS [19], we use different settings and include them in the supplementary material, as well as our MCTD approach. Benchmarks are made mainly on Google Colab with a Tesla P100 graphic card. Across all experiments, we set the number of evaluation calls to 3000.

### 5.2   Overall Performance

**Evaluation Metrics**   For each benchmark function, we run baselines and our algorithm by at least five different random seeds. Due to the limit on the computational power available to us, we set the number of calls to the objective function to 3000. Our study evaluates the best-found value at every step and computes the mean and standard deviation of all runs. As a result, we can compare

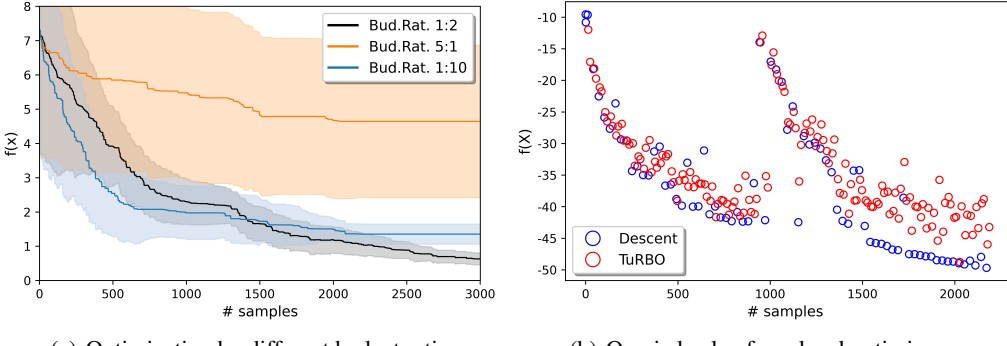

(a) Optimization by different budget ratio     (b) Queried value from local optimizers

Figure 2: (a) illustrates the optimization curves for Ackley-100d when the computational budget is divided between local descent and local BO in the ratio of 1:2, 5:1, and 1:10; (b) shows the values of Michalewicz-100d from local descent and local BO at each query.

the best-found value at the end of the run as well as the speed at which each algorithm is capable of reaching the most optimal result. There is a possibility that some algorithms will find the optimal value before 3000 calls, which will result in an early stop.

**Efficiency**    Fig. 1 illustrates the comparison between our model and baselines on benchmark sets. It was found that in general, random search, CMA, and NM methods performed poorly in these cases since they do not cooperate with any approach that may potentially improve the efficiency of the sample.

According to Fig. 1 (a), (b), and (c), MCTD significantly improves the speed of finding better results for the set of synthetic functions compared to TuRBO and La-MCTS. In particular, the Ackley synthetic function exhibits a noticeable improvement when we balance local optimization exploitation and state space exploration. Michalewicz is improved moderately through descent optimization, and MCTS helps improve the optimization consistently.

The Mujoco benchmark problems are very difficult for global optimization. Our approach is competitive with TuRBO and La-MCTS on this set and has moderate improvement over the average value on functions Hopper-33d, Walker-204d, and Cheetah-102d. In particular, the combination of local BO and local descent optimization speeds up the optimization during its early stages. It is, however, difficult to balance local exploitation and space exploration by picking the correct weights to bring recent improvements, exploration terms, and objective function values into the same order of magnitude. This is because we use the absolute value of the objective function that varies significantly at different optimization steps. In light of this, we see a large variation from different runs in this set, as in Fig.1 (d), (e), (f), and (g).

In CIFAR-10/CIFAR-100, MCTD reaches the optimal solution by a small number of samples, which is critical for NAS searches. The combination of descent and modeling approaches facilitates the search for the optimal solution more quickly than if only one method was used.

**Descent Optimizer and Bayesian Optimizer**    We examine the performance of our approach when the computational budget is divided between a local descent model and a local Bayesian optimizer TuRBO. Fig. 2(a) illustrates the optimization history of Ackley-100d when budget ratios are 1:2, 5:1, and 1:10. It is demonstrated that a model with a high budget for local descent suffers from a low optimization rate. In contrast, the model with a high budget in the local Bayesian optimizer may have difficulty escaping the local optimal point.

As shown in the case of Fig.2(a), when we use the budget that emphasizes local descent (budget 5:1), the performance is less compared with that of emphasizing local BO (budget 1:20) in term of optimization speed. Based on the budget ratio for every function in supplementary material Tab.2, it is generally advantageous to use at least the same (or even more) amount of computational budget on local BO as on local descent. This may be one challenge for the local descent approach, since this indicates that local descent may require local BO as the oracle when the function landscape is

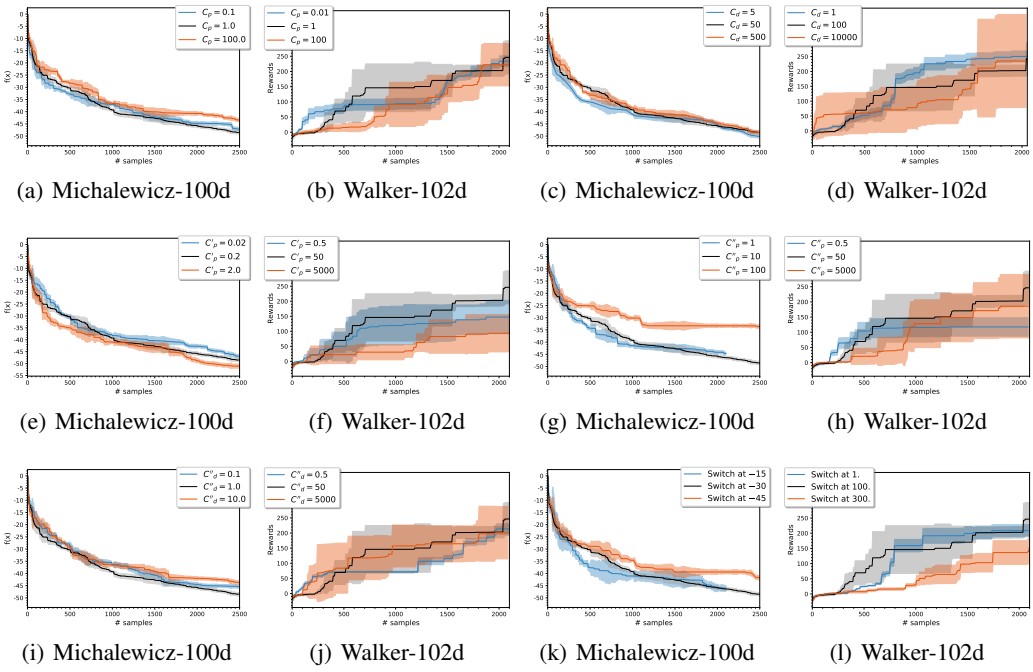

Figure 3: Ablation studies on function Michalewicz-100d with hyper parameters (a) $C_p$, (c) $C_d$, (e) $C_p'$, (g) $C_p''$, (i) $C_d''$ and (k) switching to fine-grain STP at function value; ablation studies on function Walker-102d with hyper parameters (b) $C_p$, (d) $C_d$, (f) $C_p'$, (h) $C_p''$, (j) $C_d''$ and (l) switching to fine-grain STP at function value.

difficult, such as for the complicated functions in MoJoCo locomotion and non-continuous functions in NAS sets.

However, the local descent approach still proves beneficial despite these factors. From Fig.2(b) we can see the optimization improvement of the local BO becomes insignificant when the process is close to the local optimum. Local descent, on the other hand, can contribute steadily to the discovery of a superior solution. To conclude, using a balanced approach can yield better results than using each approach separately.

## 5.3 Ablation Studies

We also perform ablation studies to understand the effect of the hyperparameters used in the algorithm, in three categories. The first category includes $C_p'$ (the weight in $uct_{exp}$) $C_p''$, and $C_d''$ (the weights on leaf exploration in Eq.8) that control the expansion of the tree. The second set of values, $C_d$ and $C_p$ in Eq.4, balance local exploitation and space exploration. Lastly, the threshold value determines when fine-grain descent is required. We use the synthetic function Michalewicz-100d and the locomotion Walker-102d for the ablation study, and each case runs with at least 3 different seeds. Please note that hyperparameters in results may be different than those presented in Section 5.2. We found that a wise choice on $C_p$, $C_p'$, and $C_p''$ is critical to improving performance, while $C_d$ and $C_d''$ are less significant. The switching threshold value is highly dependent upon the objective function's properties.

**State Space Exploration** The parameters $C_d$ and $C_p$ balance exploration and exploitation of the existing tree. As shown in Fig.3(a), the moderate choice on $C_p$ improves the overall performance slightly; however, this is not clearly observed in Fig.3(b). From Fig.3(c) and 3(d), we can see a variation in $C_d$ may not help significant changes in the overall performance. Even so, we can observe a contribution from $C_d$ and $C_p$: from Fig. 3(b) and Fig.3(d), we can see that path selection with low values of $C_d$ and $C_p$ leads to little variation between runs since the path selection tends to select the node where the current best-known value resides, thus limiting the path selection.

**Tree Expansion**    The hyperparameters $C'_p$, $C''_p$ and $C''_d$ are important for expanding the current tree. As a result of setting the parameters $C'_p$, $C''_p$ to large values and $C''_d$ to a small value, it is likely that a new sibling leaf will be created at the selected node to explore the state space. Alternatively, the path will tend to select the node that has the current optimal value. Since a leaf always has zero children, even though $C''_d$ and $C''_p$ have the same functionality as $C'_p$, the criteria for tree exploration are different for branch and leaf nodes. According to Fig.3(f) and Fig.3(h), it is evident that a good choice on $C'_p$ and $C''_p$ can improve the optimization performance by exploring new state space distant from the local optimal value. Conversely, when their values are set either too large (orange lines in Fig.3(f) and Fig.3(g)) or too small (blue line in Fig.3(h)), this would adversely affect the overall performance of the optimization process. The effect of $C''_d$ is less noticeable. However, a small $C''_d$ results in a small variation between different runs - a similar behavior as $C_d$.

**Switching at Function Value**    The fine-grain STP can be beneficial in certain cases, as the orange lines show in Fig.3(k) and Fig.3(l). In these two lines, switching takes place at a late stage of optimization, which results in excessive use of normal STP. Generally, fine-grain STP can be used as soon as possible. However, in some experiments, the fine-grain STP exploits too much in a small neighborhood at an early stage of optimization and led to low-quality GP models.

## 6    Conclusion

In this paper, we proposed novel designs for using the MCTS framework in BBO problems, with more emphasis on sample-efficient local descent, instead of using MCTS for explicit space partitioning. We design new descent methods at vertices of the search tree that incorporate stochastic search and Gaussian Processes. The local models provide surrogate objectives to propose future samples without querying the ground truth function, and they also provide uncertainty metrics for exploration steps. We propose the corresponding rules for balancing progress and uncertainty, branch selection, tree expansion, and backpropagation. We evaluated the proposed methods on challenging benchmarks and observed clear benefits in improving the efficiency of BBO methods.

## 7    Acknowledgement

This material is based on work supported by DARPA Contract No. FA8750-18-C-0092, AFOSR YIP FA9550-19-1-0041, NSF Career CCF 2047034, NSF CCF DASS 2217723, and Amazon Research Award. We appreciate the valuable feedback from Ya-Chien Chang, Chiaki Hirayama, Zhizhen Qin, Chenning Yu, Eric Yu, Hongzhan Yu, Ruipeng Zhang, and the anonymous reviewers.

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

## Appendix A    Tree expansion illustration

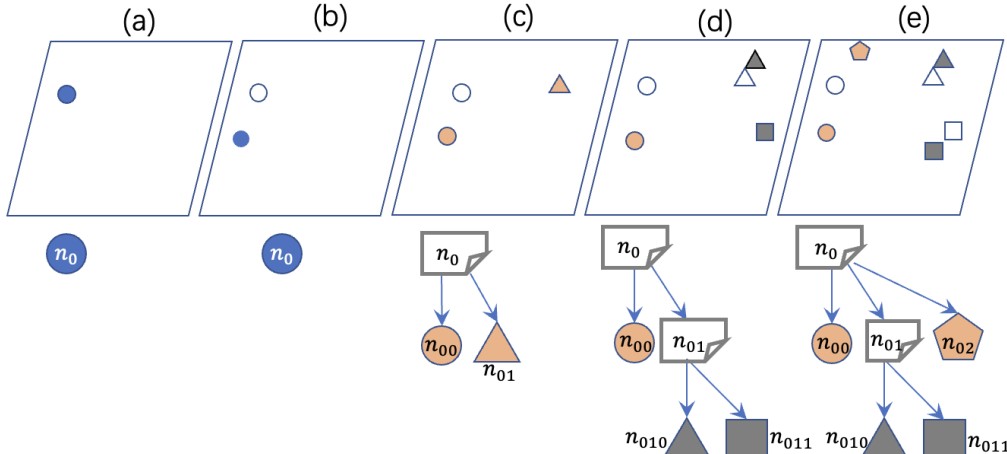

Figure 4: Top: Illustration of the nodes in the input domain; Bottom: Illustration of the tree expansion. (a) The root $n_0$ begins with a random sample. (b) Optimization is carried out on the root $n_0$. (c) Leaf exploration on the root $N^0$ creates two nodes $n_{00}$ and $n_{01}$. $n_{00}$ starts from $x_0^*$, and $n_{01}$ starts from a point distant from $x_0^*$. (d) Leaf exploration on the node $n_{01}$, generating two new node $n_{010}$ and $n_{011}$. $n_{010}$ starts from $x_{01}^*$ while $n_{011}$ starts at a point away from $x_{01}^*$. (e) Branch exploration at root $n_0$ creates a new child node $n_{02}$.

## Appendix B    Benchmark Sets

**Synthetic Functions**    We chose Ackley and Michalewicz from the synthetic function set in the nonlinear optimization benchmark [21]. Ackley is a function with multiple local minima, and Michalewicz has steep valleys and many ridges. We use Ackley-50d, Ackley-100d, and Michalewicz-100d as our benchmark.

**MuJoCo Locomotion**    For reinforcement learning problems from MuJoCo locomotion environments [22], we chose Hopper, Walker, and HalfCheetah for tests. Hopper has 3 dimensions in action space $a$ and 11 in observation $s$. We choose a linear policy $a = Ws$ in which $W$ is the weighting matrix to search for maximizing the reward, therefore, the search space for Hopper-33d is in the dimension of $3 \cdot 11 = 33$. Similarly, we set linear policies in both Walker-102d and HalfCheetah-102d. In addition to the above linear policy, we double the weighting matrix space dimension in Walker from 102 to 204, such that $a = W_1 s + W_2 s$ where $W_1$ and $W_2$ are matrices in the dimension of 102. In this case, the optimization problem is Walker-204d. Since our approach considers deterministic results, we set the noise scale to zero in all MuJoCo environments to avoid randomness in rewards.

**Neural Architecture Search**    For the NAS benchmark, we use two datasets CIFAR-10 and CIFAR-100 from NAS-Bench-201 [23]. Each network in the datasets consists of three stacks of searching cells, and each cell has six positions where one can select one type of layer from five different types: (1) zeroize, (2) skip connection, (3) 1-by-1 convolution, (4) 3-by-3 convolution, and (5) 3-by-3 average pooling layer. Overall, there are $5^6 = 15625$ different types of architectures, and each architecture is trained and evaluated on both CIFAR-10 and CIFAR-100. The accuracy of training and evaluation is recorded. To benchmark this set, we created the following functions in the real domain: we replace each of the five types of layers with an integer, and the real-valued input is rounded up to the nearest integer. The evaluation accuracy of the architecture is set as the function value. As an example, we set the input domain to $\{[0.5, 5.5]^6\}$, and $f([1.1]^6) = f([1]^6)$, where each $1 - 5$ corresponds to one type of the layers. It should be noted that in this method different inputs may refer to the same network architecture; therefore, the number of unique architectures examined is less than the number of functions called.

