# OpenReview forum: "Monte Carlo Tree Descent for Black-Box Optimization"
_NeurIPS.cc/2022/Conference — NeurIPS 2022 Accept_

### Official Review · Reviewer_sxfi · 2022-07-11

**Rating:** 6
**Confidence:** 4
**Soundness:** 2 fair
**Presentation:** 2 fair
**Contribution:** 2 fair

**Summary:**

This paper proposes a new MCTS-based optimization method that combines descent methods and BO. Different from LA-MCTS, the proposed MCDescent does not use partition due to the limitation of high-dimensional space and uses a set of unique mechanisms in the MCTS framework. Expeirments show the MCDescent can outperform baselines and sota black-box optimization methods.

**Questions:**

How does MCDescent compare to other methods in terms of running time? The author should discuss this and shows it in the experiments.

**Limitations:**

Yes.

**Strengths And Weaknesses:**

Although the experimental results show the superior performance of the proposed method, I have the following concerns.

1. The novelty of the proposed method is limited. Authors may need to highlight the motivation and make it reasonable, explain in detail why you combine the two approaches.

2. The writing of this paper should be improved. For example,  Algorithm 1 should be rewritten for clarity, the notations used in the current version are very confusing and inconsistent.

3. More experiments should be done to test the sensitivity of the hyper-parameters in the MCDescent, e.g., $C_p$.

4. The number of samples is too small for the RL tasks. Although the authors want to examine the performance of different algorithms under a limited number of evaluations, the too-small budgets lead to inferior performance, and the policy does not converge on some tasks. For example, in the Walker environment, the number of samples used in LA-MCTS paper is 40000. However, it is only 3000 in this paper.

5. The authors said, " the performance of MCDescent may be limited in some high dimensional spaces that are inhospitable to descent
approaches". It is better to examine MCDescent on high-dimensional synthetic functions.

There are also some minor problems.

1. More references are needed, such as lines 34-35.

2. The experiments of Tree expansion and Optimization route can not be named "Ablation study."

---

> ### Author Response · Authors · 2022-08-02
> **Motivation highlighted and parameter study is appended to supplementary material**
>
> Thank you for your feedback. We address the specific questions as follows:
>
> >  Authors may need to highlight the motivation and make it reasonable, explain in detail why you combine the two approaches.
>
> It is a common practice to run multiple optimization processes with different initial conditions in black-box optimization. A key part of our idea is to integrate all optimization processes into a tree structure, and instead of running each process to its end, we would prefer to select the most promising process at each stage. On the basis of this concept, we construct the MCTS structure incorporating all of these processes.
> Furthermore, the combination of both modeling and direct search approaches can be beneficial.
> In most modeling approaches, a sample is selected randomly in a trusted region where the information is collected.
> Surrogate models may, however, utilize historical data from search models rather than randomly selecting samples from the neighborhood. Consequently, one sample that is used in the search model to determine the descent direction also provides information for constructing the surrogate model. In direct search optimization, since the descent direction of a local point is approximated from the learning model, the search is not completely random. As a backup strategy, the STO proposes an alternative search in the opposite direction in the event that the first search fails to yield a decent result. In this way, the sample efficiency can be improved by combining the two different approaches.
>
> > Algorithm 1 should be rewritten for clarity.
>
>  I have rewritten the algorithm 1, and I would like to clarify three equations:
> The equation of UCT in path selection (as in line 5 of Algorithm 1 and equation (2) in main text) is
>
> $$
> uct_i  = -y^*_{i}        + C_d \cdot  \sum_{j=1}^{J} (dy_{i,-j})       + C_p \cdot  \sqrt{\log{n_{node}}/n_{i}}
> $$
>
> Here, $C_d$ is a weight factor controlling the importance of recent improvements, $C_p$ is a hyper-parameter for the extent of exploration, $n_{node}$ and $n_i$  are the number of visits to the parent node and child $i$, respectively. $y*$ i is the best value found at the child node $i$, $dy_{i, -j}$ is the most recent j’s improvement at node $i$, ($j=1,...,J$), and is set to 0 if no improvement is obtained at that step.
>
> The equation on checking if a new branch should be added ((as in line 6 of Algorithm 1 and line 195 in main text) is:
>
> $$
> UCT_{explore} = -\sum_{i=1}^{N} (y^*_{i})/N+C'_p \cdot \sqrt{  \log{n_n}   }
> $$
>
> In which $C_p$' is also a hyper-parameter for the extent of exploration, $N$ is the number of children to the checking parent node, and $n_n$ is $n_{node}$ as above equation (There is an error at displaying $n_{node}$ in above line.)
>
> The equation on determining if a leaf node is worth exploitation:
>
> $$
> -y^*_{i}        + C_d \cdot  \sum_{j=1}^{J} (dy_{i,-j})   > C''_p \cdot n_f
> $$
>
> In which  $C''_p$  is another parameter for the extent of exploration, and $n_f$ is the number of visits on leaf node.
>
> >More experiments should be done to test the sensitivity of the hyper-parameters in the MCDescent,
>
> We have made an ablation study in Supplementary material, including $C_d$, $C_p$, $C'_p$, $C''_d$, $C''_p$, and function value to switching to fine-grained STO.
>
> >The number of samples is too small for the RL tasks.
>
> We are limited by the hardware resources available to us.  Models based on BO are memory-intensive.
> We ran several TuRBO/LaMCTS runs up to 10K samples on an HPC node with 256GB memory, but the resources were unavailable at this time.
> MCDecent runs on Google Colab, with only 16GB of memory (DRAM and GPU), and cannot accommodate more than 3.5K samples.
> Moreover, the average reward for LaMCTS at 10K samples is 392, and at 3K it is 297, which accounts for 75% of the rewards from 10K samples. Therefore, we usually limit the comparison to 3K.
>
>
> > It is better to examine MCDescent on high-dimensional synthetic functions.
>
> In supplementary material we also added a performance plot for "Ackley-500d" which has many local minima. The budget ratio (descent : BO) is set to 1:1 which is the same value for Ackley-50d and Ackley-100d. We figured out that the contribution from STP model is greated reduced on this function compared with Ackley-50d and Ackley-100d. In order to achieve better performance, we have to raise the computational power spent on BO.
>
> > How does MCDescent compare to other methods in terms of running time?
>
> Thank you for pointing this out.
> Nevertheless, due to limited resources, we conduct our testing on different types of machines.
> For the function Michalewicz-100d:
> * Nelder-mead and CMA can process 3K samples in a matter of minutes.
> * It may take TuRBO 2 hours for first 3K samples and LaMCTS 6 hours for first 3K samples on a HPC node (AMD EPYC 7742, 4608 GFlop/s, 256 GB of DDR4 RAM)
> * The MCDecent collects 3K samples within two hours on a Google Colab platform (Intel(R) Xeon(R) 2.20GHz, 16GB DRAM, P100 GPU)

---

> > ### Comment · Reviewer_sxfi · 2022-08-08
> > **I'm increasing my score**
> >
> > After reading the responses by the authors, several issues have been addressed.
> > However, I still think the motivation of this paper is not convincing enough for me.
> > Due to the sufficient experimental results, I'm increasing my score to weak accept.

---

### Official Review · Reviewer_6Fro · 2022-07-11

**Rating:** 2
**Confidence:** 5
**Soundness:** 1 poor
**Presentation:** 1 poor
**Contribution:** 1 poor

**Summary:**

The paper considers optimization of deterministic black-box functions. The proposed algorithm combines Monte Carlo tree search with local descent/bayesian optimization. Empirical work shows elicits strong performance of the proposed method compared to various baselines on a diverse set of domains.

**Questions:**

-

**Limitations:**

-

**Strengths And Weaknesses:**

The article is poorly written, and often it is difficult to figure out the intention of the authors. For instance, in Algorithm 1, the index i is undefined and probably is different from the index i in (2), which probably is j in Alg.1. dy is also undefined in the algorithm, although (2) gives sufficient hint about it. Other variables are undefined as well, but familiarity with MCTS is sufficient to have a reasonable guess. The authors often inform us what the algorithm is not, instead of focusing on a clear description of the algorithm, and explaining the differences subsequently.

Therefore, while the algorithm could be interesting, it is fairly difficult to evaluate the strength of it.

The empirical results seem to indicate that the proposed algorithm performs well. Some of the domain choices seem a bit strange for black-box optimization (why would we want to optimize a linear policy on a MuJoCo domain this way, and not a more complex policy by reinforcement learning?), but ultimately it is possible to use these domains as well for testing.

-------------------------
After rebuttal:

I appreciate the effort of the authors to improve the paper according to the suggestions of the reviewers. I hope, it will become a better paper after a few more iterations.

---

> ### Author Response · Authors · 2022-08-02
> **Clarify the concept and key equations**
>
> Thank you for your feedback. We address the specific questions as follows:
>
> > The article is poorly written, and often it is difficult to figure out the intention of the authors. For instance, in Algorithm 1, the index i is undefined and probably is different from the index i in (2), which probably is j in Alg.1. dy is also undefined in the algorithm, although (2) gives sufficient hint about it. Other variables are undefined as well, but familiarity with MCTS is sufficient to have a reasonable guess. The authors often inform us what the algorithm is not, instead of focusing on a clear description of the algorithm, and explaining the differences subsequently.
>
> It is a common practice to run multiple optimization processes with different initial conditions in black-box optimization. A key part of our idea is to integrate all optimization processes into a tree structure, and instead of running each process to its end, we would prefer to select the most promising process at each stage. On the basis of this concept, we construct the MCTS structure incorporating all of these processes.
>
> Furthermore, we believe that the combination of both modeling and direct search approaches can be beneficial.
> In most modeling approaches, a sample is selected randomly in a trusted region where the information is collected.
> Surrogate models may, however, utilize historical data from search models rather than randomly selecting samples from the neighborhood. Consequently, one sample that is used in the search model to determine the descent direction also provides information for constructing the surrogate model. The efficiency of the sample can be improved by utilizing the same sample in two different approaches.
>
> In direct search optimization, since the descent direction of a local point is approximated from the learning model, the search is not completely random. As a backup strategy, the STO proposes an alternative search in the opposite direction in the event that the first search fails to yield a decent result.
>
>
>
> Additionally, I would like to clarify three equations:
> The equation of UCT in path selection (as in line 5 of Algorithm 1 and equation (2) in main text) is
>
> $$
> uct_i  = -y^*_{i}        + C_d \cdot  \sum_{j=1}^{J} (dy_{i,-j})       + C_p \cdot  \sqrt{\log{n_{node}}/n_{i}}
> $$
>
> Here, $C_d$ is a weight factor controlling the importance of recent improvements, $C_p$ is a hyper-parameter for the extent of exploration, $n_{node}$ and $n_i$  are the number of visits to the parent node and child $i$, respectively. $y*$ i is the best value found at the child node $i$, $dy_{i, -j}$ is the most recent j’s improvement at node $i$, ($j=1,...,J$), and is set to 0 if no improvement is obtained at that step.
>
> The equation on checking if a new branch should be added ((as in line 6 of Algorithm 1 and line 195 in main text) is:
>
> $$
> UCT_{explore} = -\sum_{i=1}^{N} (y^*_{i})/N+C'_p \cdot \sqrt{  \log{n_n}   }
> $$
>
> In which $C_p$' is also a hyper-parameter for the extent of exploration, $N$ is the number of children to the checking parent node, and $n_n$ is $n_{node}$ as above equation (There is an error at displaying $n_{node}$ in above line.)
>
> The equation on determining if a leaf node is worth exploitation:
>
> $$
> -y^*_{i}        + C_d \cdot  \sum_{j=1}^{J} (dy_{i,-j})   > C''_p \cdot n_f
> $$
>
> In which  $C''_p$  is another parameter for the extent of exploration, and $n_f$ is the number of visits on leaf node.

---

### Official Review · Reviewer_z8ZA · 2022-07-11

**Rating:** 7
**Confidence:** 2
**Soundness:** 4 excellent
**Presentation:** 3 good
**Contribution:** 3 good

**Summary:**

This paper focuses on the problem of Black-Box-Optimization, optimizing a function while only being able to sample its value at different points.  The authors propose a novel combination of building a tree to organize different regions that are being searched, and uses modifications of STP and TuRBO to optimize from a given point.  The entire method balances exploring new regions for more promising spots with exploiting and optimizing in a good spot.  Modified UCT is used to guide traversal of the tree, which also includes an option to create a new child at each node.  This overall method is experimentally evaluated in many settings and shown to be competitive with previous approaches, while surpassing their performance on many problems.


**Questions:**

- On line 267 you state that some high dimensional spaces are inhospitable to descent approachs?  What are the characteristics of these inhospitable spaces? How can you tell if a space is inhospitable?


**Limitations:**

This was addressed to my satisfaction.


**Strengths And Weaknesses:**

Strengths:
- I quite liked this paper and felt that its approach to combining MCTS with BO was unique and makes a very solid contribution.
- The evaluation is thorough and complete and is very convincing as to the strengths of the proposed methods.
- I appreciated the ablation studies that were done.  Overall the evaluation feels solid.
- The paper is well-written and clear.

Weaknesses:
- The fact that the proposed methods are focused on deterministic functions wasn't made explicit until Line 216 on page 6, and later in the conclusion.  I think this would have been good to make clear from the very beginning.
- The uncertainty regions in some of the results are very large.  Could more runs have been performed to bring down this uncertainty?
- I am not sure that Figure 4 conveyed much meaning to me.  It was hard to understand the point you were trying to get across.

Minor feedback:
- Line 67-68 - This sentence doesn't make very much sense to me as written.
- Line 122 - "... have similar values..." might better be: "... have more similar values..."
- Algorithm 1 - "GP.correlation lengh" seems like it should be "GP.correlation length"
- Line 176 - "In the scene the ..." might sound better as "In the event that the ..."
- Line 227 - "...we create following ..." perhaps should be "...we create the following..."

---

> ### Author Response · Authors · 2022-08-02
> **Direct search in high dimension space experiment added**
>
> Thank you for your careful reading of our paper. We answer the specific questions as follows.
>
> >The fact that the proposed methods are focused on deterministic functions wasn't made explicit until Line 216 on page 6, and later in the conclusion. I think this would have been good to make clear from the very beginning.
>
> Thank you for your suggestion. We will add this at the beginning of the text.
>
> > The uncertainty regions in some of the results are very large. Could more runs have been performed to bring down this uncertainty?
>
> I suppose the result with “Walker-204d” is the one with large uncertainty, as in all others our approach has similar uncertainty as TurBO and LaMCTS.
> We have added additional runs to it, and updated the plot in main pdf as well as put it into the supplementary material.
>
>
> > I am not sure that Figure 4 conveyed much meaning to me. It was hard to understand the point you were trying to get across.
>
> Our study on Fig 4. is to examine the optimization trajectory on the objective function landscape and see how the trajectories differ from each method.
> In BO-based TuRBO models, the optimization begins by sampling randomly within the trusted region, and gradually locates the optimal point; in such a case, the trajectory swirls around to collect information about the state space.
> La-MCTS utilizes TuRBO and emphasizes the exploration of the entire state space. As we can see, such a trajectory will explore a larger area of space. With our MCDescent, we utilize TuRBO, but use direct search to locate the optimal point in a shorter period of time.
>
>
> > Line 67-68 - This sentence doesn't make very much sense to me as written.
>
> Thank you for your careful reading. Here I would like to express it as: “Bayesian Optimization algorithms are a typical class of algorithms that uses modeling approaches. These algorithms build their surrogate models based on the Gaussian Process.”
>
>
> > On line 267 you state that some high dimensional spaces are inhospitable to descent approaches? What are the characteristics of these inhospitable spaces? How can you tell if a space is inhospitable?
>
> Direct search algorithms often suffer from highly non-smooth objectives with many local minima. Despite the fact that the number of  iterations required to produce an error  (with $E[f(x) - f(x^*)] < \epsilon$ )  is proportional to the number of dimensions n ( with a complexity of $O(n)$, [Bergou, et al.]), we wish to improve its performance in order to enable the STP method to find a significant descent direction and step with a limited number of steps. Typically, such work in high dimensional space requires the use of a small initial step size. However, small initial steps in descent lead to samples that are close to one another, which is not preferred to BO at the initial stage.
>
> In supplementary material we also added a performance plot for "Ackley-500d" which has many local minima. The budget ratio (descent : BO) is set to 1:1 which is the same value for Ackley-50d and Ackley-100d. We figured out that the contribution from STP model is greated reduced on this function compared with Ackley-50d and Ackley-100d. In order to achieve better performance, we have to raise the computational power spent on BO.
>
> [Bergou, E. H., Gorbunov, E., and Richtárik, P. Stochastic three points method for smooth minimization.
> SIAM Journal on Optimization, 30(4):2726–2749, 20]

---

### Official Review · Reviewer_TrTs · 2022-07-12

**Rating:** 6
**Confidence:** 4
**Soundness:** 3 good
**Presentation:** 3 good
**Contribution:** 2 fair

**Summary:**

This paper proposes a method for Black-Box Optimization which incorporates the idea of UCT in sampling.
The proposed method, MCDescent, shows promising results in several benchmarks.


**Questions:**

There should be a discussion focusing on the hyperparameters.
For example, if possible, I would like to see the following explanation.
1. How sensitive the performances are to the hyperparameters.
1. How the authors chose the values in Table 2.
1. Is there any guideline for tuning the hyperparameters?

Where is the definition of $C_d$?
I may have overlooked it, but I could not find it in the main material.
(It is mentioned in the supplementary as "weight of recent improvement.")

I think there are better citations for CMA-ES.
Please explain the reason if it is meaningful to cite this specific implementation.

This is not a question, but I would like to see Figure 1 of the appendix
in the main material if possible because it is very helpful for understanding the algorithm.

**Limitations:**

I could not think of any direct possibility of potential negative societal impact.


**Strengths And Weaknesses:**

Strengths:
The idea is simple, and the algorithm is easy to understand.
Probably the implementation is also not complex.

Weaknesses:
The algorithm has eight hyperparameters, and their values vary for different benchmarks.
Also, there is no explanation about how they are tuned.
I would be happy to give the paper a better score if this weakness is solved.

---
Post rebuttal.
Thank you very much for the detailed comments on my concerns.
I changed my score.

---

> ### Author Response · Authors · 2022-08-02
> **Hyper-parameter study added**
>
> Thank you for your comments and questions. We have inlined our responses.
>
> > Weaknesses: The algorithm has eight hyperparameters, and their values vary for different benchmarks. Also, there is no explanation about how they are tuned. I would be happy to give the paper a better score if this weakness is solved.
>
> Reply: In our approach, we emphasize the absolute value of the objective function at every step of the path selection and tree exploration stages. As a consequence, we will have to set up these hyper-parameters for tuning in order to fit the scaleness of the function value and the optimization improvement. Here is how they were tuned in our approach.
> The budget ratios, initial step size $\alpha$, and switching value for descent optimization are determined by running BO and descent optimization only on the root node. It is recommended to switch to the fine-grained model when the performance of STP becomes limited. If, however, the STP outperforms the BO significantly, the budget for descent optimization is reduced.
> $C_d$, $C''_d$ and $C_p$ should be set to reflect the acceptable improvement speed of the optimization on a single node and should be adapted to the same scaleness of the function value.
> The weighting factors $C'_p$ and $C''_p$ for tree exploration are set in such a way to ensure that the exploitation on a node is not "optimization saturated".
>
> We also made an ablation study on hyper parameters appended to Supplementary material.
> $C_d$ emphasizes the recent improvement. A small $C_d$ will make the path focus on the node with best value at the current step and may lead to local minima. However, a large $C_d$ will over emphasize the recent improvement, so the path tends to focus on newly added nodes due to their large improvement during the first few steps. $C_p$ and $C'_p$ control the selection of branches. When they are high, sub-optimal nodes or new branching nodes are selected for exploration.
>
> $C''_p$ and $C''_d$ are responsible for splitting the leaf node. When these two parameters are set to large values, it is likely that a new sibling leaf will be created at the selected leaf, which will significantly reduce the performance of the algorithm. In contrast, A path with small values will tend to select the node with a local optimal value. In spite of the fact that $C''_d$ and $C''_p$ have the same functionality as $C_d$, $C_p$, and $C'_p$, the criteria for tree exploration at leaf node is different since a leaf always has zero children, which means the normal UCT formula is no longer valid.
>
> > Where is the definition of $C_d$ I may have overlooked it, but I could not find it in the main material. (It is mentioned in the supplementary as "weight of recent improvement.")
>
> Here, I would like to clarify three equations:
> The equation of UCT in path selection (as in line 5 of Algorithm 1 and equation (2) in main text) is
>
> $$
> uct_i  = -y^*_{i}        + C_d \cdot  \sum_{j=1}^{J} (dy_{i,-j})       + C_p \cdot  \sqrt{\log{n_{node}}/n_{i}}
> $$
>
> Here, $C_d$ is a weight factor controlling the importance of recent improvements, $C_p$ is a hyper-parameter for the extent of exploration, $n_{node}$ and $n_i$  are the number of visits to the parent node and child $i$, respectively. $y*$ i is the best value found at the child node $i$, $dy_{i, -j}$ is the most recent j’s improvement at node $i$, ($j=1,...,J$), and is set to 0 if no improvement is obtained at that step.
>
> The equation on checking if a new branch should be added ((as in line 6 of Algorithm 1 and line 195 in main text) is:
>
> $$
> UCT_{explore} = -\sum_{i=1}^{N} (y^*_{i})/N+C'_p \cdot \sqrt{  \log{n_n}   }
> $$
>
> In which $C_p$' is also a hyper-parameter for the extent of exploration, $N$ is the number of children to the checking parent node, and $n_n$ is $n_{node}$ as above equation (There is an error at displaying $n_{node}$ in above line.)
>
> The equation on determining if a leaf node is worth exploitation:
>
> $$
> -y^*_{i}        + C_d \cdot  \sum_{j=1}^{J} (dy_{i,-j})   > C''_p \cdot n_f
> $$
>
> In which  $C''_p$  is another parameter for the extent of exploration, and $n_f$ is the number of visits on leaf node.
>
>
> > I think there are better citations for CMA-ES. Please explain the reason if it is meaningful to cite this specific implementation.
>
> Thank you for pointing this out. There are two sources with the citation information: a Github repository at [Hansen, et. al. (2019)], and the other [Hansen, et. al. (2022)]. Both of them have the latest implementations with version r3.2.2. As I checked out the information from Github, I used the citation from there.
>
> > This is not a question, but I would like to see Figure 1 of the appendix in the main material if possible because it is very helpful for understanding the algorithm.
>
> Thank you a lot for the suggestion. This figure will be moved to the main text.
>
> [Hansen, et. al. (2022)] Nikolaus Hansen, et al. (2022). CMA-ES/pycma: r3.2.2 (r3.2.2). Zenodo. https://doi.org/10.5281/zenodo.6370326

---

### Meta-Review · Area_Chair_Jz5v · 2022-08-24

**Recommendation:** Accept
**Confidence:** Less certain

**Metareview:**

This paper proposes a novel combination of Bayesian optimization and Monte Carlo Tree Search for more sample-efficient black-box optimization. The method is adding complexity, but the empirical results are thorough and show a clear benefit.

The reviewers on this paper did not come to unanimous decision, but the clear majority advocated for acceptance, and I concur, especially because reviewer 6Fro did not spell out their concerns with sufficient concreteness.

**Award:**

No

---

### Decision · Program_Chairs · 2022-09-14

Accept